# Quality Enhancement of MPEG-H 3DA Binaural Rendering Using a Spectral Compensation Technique

**Hyeongi Moon** [1] and **Young-cheol Park** [2,*]

1 Department of Electrical and Electronic Engineering, Yonsei University, Seoul 03722, Korea; orexis@dsp.yonsei.ac.kr
2 Division of Software, Yonsei University, Wonju 26426, Korea
* Correspondence: young00@yonsei.ac.kr; Tel.: +8233-760-2756

**Abstract:** The latest MPEG standard, MPEG-H 3D Audio, employs the virtual loudspeaker rendering (VLR) technique to support virtual reality (VR) and augmented reality (AR). During the rendering, the binaural downmixing of channel signals often induces the so-called comb filter effect, an undesirable spectral artifact, due to the phase difference between the binaural filters. In this paper, we propose an efficient algorithm that can mitigate such spectral artifacts. The proposed algorithm performs spectral compensation in both the panning gain and downmix signal domains depending on the frequency range. In the low-frequency bands where a band has a wider bandwidth than the critical-frequency scale, panning gains are directly compensated. In the high-frequency bands, where a band has a narrower bandwidth than the critical-frequency scale, a signal compensation similar to the active downmix is performed. As a result, the proposed algorithm optimizes the performance and the complexity within MPEG-H 3DA framework. By implementing the algorithm on MPEG-H 3DA BR, we verify that the additional computation complexity is minor. We also show that the proposed algorithm improves the subjective quality of MPEG-H 3DA BR significantly.

**Keywords:** MPEG-H 3D Audio; binaural rendering; amplitude panning; active downmix

## 1. Introduction

The MPEG-H 3D Audio (3DA) standard [1] aims to provide immersive 3D audio for high-resolution UHDTVs. It comprises state-of-the-art technologies that support high-efficiency compression and transmission of channel, object-based, SAOC (spatial audio object coding) and HoA (high-order ambisonics) audio formats, and high-quality rendering in various layouts. Recently, the MPEG-H 3DA system was adopted in TV broadcast standards, e.g., ATSC 3.0 [2] and DVB [3], and in the virtual reality (VR) profile of mobile standard 3GPP [4].

The MPEG-H decoder can render each audio format on various playback layouts using the format converter, object renderer, SAOC renderer, and HoA renderer. In addition, the standard includes a low-complexity high-quality binaural renderer in response to the increasing use cases of mobile devices. Another important feature of MPEG-H 3DA is interactivity, i.e., the listener can manipulate each audio scene to the extent allowed by the content creator. A typical use case is to render audio based on three degrees of freedom of the head (i.e., yaw, pitch, and roll rotation). With these advantages, the MPEG-H 3DA decoder low complexity (LC) profile was selected as a 3DoF (Degrees of Freedom)/3DoF + rendering technology for MPEG-I audio standards for augmented reality (AR) and VR.

MPEG-H 3DA performs 3DoF rendering of object and channel signals [1], using virtual loudspeaker rendering (VLR) [5]. In the VLR, object signals are first converted to virtual channel (loudspeaker) signals using panning techniques, such as 3D vector basic amplitude panning (VBAP) [6] and multidirectional amplitude panning (MDAP) [7]. The virtual channel signals are filtered with low-resolution binaural room transfer function (BRTF) or head-related transfer function (HRTF) and then downmixed to generate the binaural

signal. During this binaural downmixing process, various speaker-to-ear transfer functions corresponding to different acoustics paths are mixed, resulting in the so-called comb filter effect [8]. The primary artifacts of the comb filter effect are spectral coloration and volume degradation, both of which cause inaccurate spatial image position, often significantly [8,9]. Therefore, to improve the audio quality of MPEG-H 3DA binaural rendering (BR), it is crucial to prevent the comb filter effect.

For the channel signals, the MPEG-H 3DA decoder performs BR after 'objectizing' channel signals (by regarding channel signals as object signals at the loudspeaker positions). In general, high-quality channel sources include 'artistic intentions', such as EQ, which may prevent the comb filter artifacts. However, 'artistic intents' alone cannot overcome the comb filter effect in every case of 3DoF rendering of channel sources.

This paper presents a rigorous method to improve the sound quality of MPEG-H 3DA BR. We first propose an efficient gain normalization algorithm that can compensate for the spectral artifacts caused by the comb filter effect. The previous studies on this issue mainly focused on the panning algorithm. In [7,10], the panning gain ratio was adjusted to improve the sound image localization performance. In [8,9], a panning gain normalization was used to reduce coloration. However, these methods were developed assuming loudspeaker-based listening environments. In such a case, binaural transfer functions are not considered precisely; hence they only partially prevent the comb filter effect when used for VLR. A baseline solution for this spectral coloration and distortion is active downmixing [11]. Unfortunately, active downmixing does not work when more than one sound object resides in a single processing band, which is common in many audio signals.

The proposed algorithm performs spectral compensation in both the panning gain and downmix signal domains. In order to implement frequency dependent compensation on the standard effectively, the MPEG-H 3DA frequency domain object renderer and binaural renderer working in the 64-band complex quadrature mirror filter (QMF) [12] are used. Panning gain compensation is performed in the low-frequency band where the bandwidth of the QMF is wider than the critical frequency scale. And a signal compensation similar to the active downmix is performed in the high-frequency band, where the QMF band has a narrower bandwidth than the critical frequency scale. In such a way, the proposed algorithm compromises the algorithm's complexity with performance within a framework of MPEG-H 3DA.

This paper is organized as follows. Section 2 reviews the MPEG-H 3DA decoder and BR. Section 3 demonstrates spectral artifacts in the MPEG-H 3DA BR, and Section 4 proposes a binaural gain normalization and its implementation structure on MPEG-H 3DA BR as well as an empirical analysis based on our implementation. Section 5 presents experimental results that include both objective and subjective tests. In Section 6, we present the conclusion.

## 2. Spectral Distortions in the MPEG-H 3DA BR

Figure 1 illustrates the core decoding and rendering parts of the MPEG-H 3DA. Dynamic range compression (DRC) and the HoA rendering block are omitted in the figure for brevity. The core decoder of MPEG-H 3DA converts a compressed bit-stream to a waveform (e.g., channel-based object-based audio) and associated metadata (e.g., positions of objects and a loudspeaker geometry). During the rendering stage, the associated data are processed by the scene-displacement interface. It calculates and applies the rotation matrix to update the positions of audio objects and channels, both of which depend on user interaction (e.g., user's yaw, pitch, and roll movement). Using 3D VBAP, the object renderer then takes the object and channel signals and updated positions to produce virtual loudspeaker signals. Finally, the binaural renderer generates a binaural signal by filtering the channel signal by BRTFs/HRTFs that corresponds to the virtual loudspeaker position [13].

Figure 2 shows more detailed schematics of the VLR-based BR in MPEG-H 3DA. For a given loudspeaker layout and head rotation information, the VLR system generates channel

signals for all object signals. Following 3D VBAP, the $i$-th channel signal $X_i(k)$ for input object signals $S_m(l,k)$, $m = 1, 2, ..., N_{obj}$, is obtained as

$$X_i(l,k) = \sum_{m=1}^{N_{obj}} \frac{g_{i,m}}{\|\mathbf{g}_m\|_p} S_m(l,k) = \sum_{m=1}^{N_{obj}} \tilde{g}_{i,m} S_m(k), \ \ i = 1, 2, ..., N_{ch}, \tag{1}$$

where $l$ and $k$ are the frame and frequency indices, respectively, and $\|\mathbf{g}_m\|_p$ represents the $p$ norm of the panning gain vector $\mathbf{g}_m = [g_{1,m}, g_{2,m}, ..., g_{N_{ch},m}]^T$. $N_{obj}$, and $N_{ch}$ denotes the total numbers of input sound objects and virtual loudspeakers.

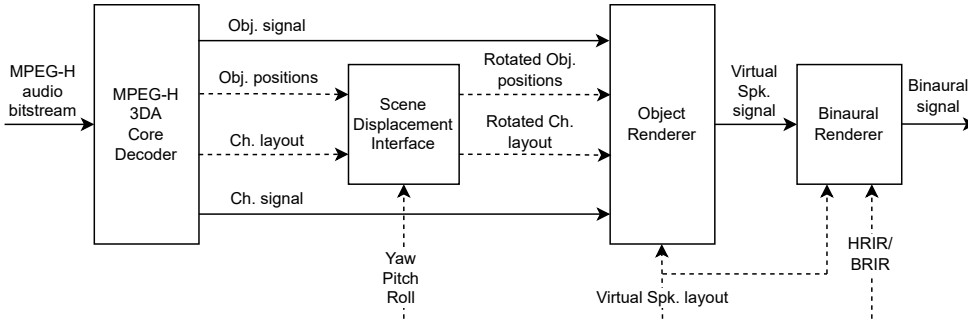

**Figure 1.** A block diagram of the MPEG-H 3DA 3DoF rendering of audio objects and channel signals.

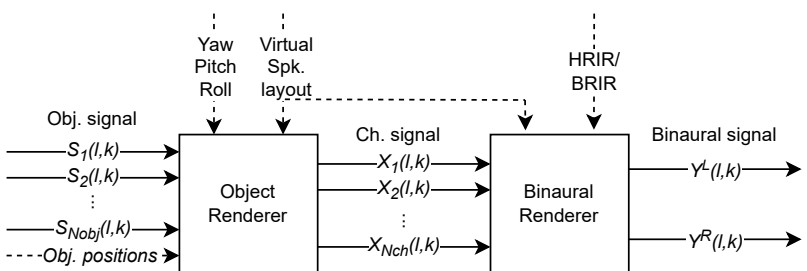

**Figure 2.** Block diagram of the VLR-based BR in MPEG-H 3DA.

The normalized panning gain $\tilde{g}_{i,m}(= g_{i,m}/\|\mathbf{g}_m\|_p)$ allows the maintenance of a constant loudness regardless of the panning direction. The norm order $p$ can be either 1 or 2 depending on the coherence between the binaural filters [8]. The ear signals are then obtained by filtering the channel signal with BRTFs/HRTFs that correspond to the virtual loudspeaker location. The binaural signal $Y^j(l,k)$ in Figure 2 is obtained by summing the ear signals, as follows, where the frequency-transformed BRTFs/HRTFs are denoted as $H_i^j(k)$, $j \in \{L, R\}$.

$$Y^j(l,k) = \sum_{i=1}^{N_{ch}} X_i(l,k) H_i^j(k), \ \ j \in \{L, R\}. \tag{2}$$

The transfer function of the MPEG-H 3DA BR system between the $m$-th sound object to the binaural signal, defined as $H_{BR}^j(k) = Y^j(k)/S_m(k)$, can be expressed as

$$H_{BR}^j(k) = \sum_{i=1}^{N_{ch}} \tilde{g}_i H_i^j(k) = \tilde{\mathbf{g}}^T \mathbf{H}^j(k), j \in \{L, R\}, \tag{3}$$

where $\tilde{\mathbf{g}} = [\tilde{g}_1, ..., \tilde{g}_{N_{ch}}]^T$ and $\mathbf{H}^j(k) = [H_1^j(k), ..., H_{N_{ch}}^j(k)]^T$. Here, for simplicity, we omit the frame index $l$ and assume $m = 1$.

The comb filter effect occurs during the summing of the ear signals as Equation (2). The effect can be illustrated by calculating the transfer function between an object signal and the left ear signal, i.e., $|H_{BR}^L|$, following the standard of MPEG-H 3DA BR [1], as in

Figure 3. During the calculation, we assume that a source at an angle of 10° to the left from the frontal direction is rendered using a pair of virtual loudspeakers located at 30° and 0°, respectively. The diffuse-field equalized (DFE) MIT HRTF [14] is used as a virtual loudspeaker, and $p$ is set to 2 for the gain normalization.

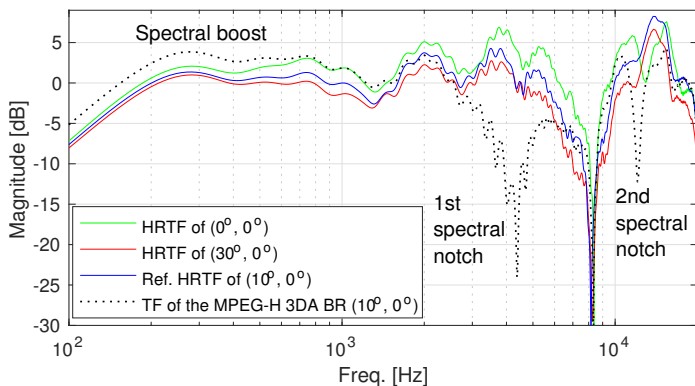

**Figure 3.** $|H^L_{BR}(k)|$ (dotted line) for a virtual source at $+10°$ to the right from the frontal direction, rendered using a pair of loudspeakers at 30° and 0° ($p = 2$). Left ear is located at the azimuth of 90°.

The transfer function in Figure 3, compared to true HRTFs, shows two main artifacts: broad spectral notches near 3.3 kHz as well as 12 kHz and a broad spectral boost below 1.5 kHz. These distortions are commonly observed in practical loudspeaker setups and directly affect the perceptual quality of the downmixed sound.

A simple solution is active downmixing [11] that compensates for the spectral distortions during downmixing of the ear signals. However, the active downmixing applies gain to a downmixed binaural signal rather than an individual object signal. Consequently, some unwanted spectral distortion may occur during the step if multiple objects are in the same processing band. Therefore, a rigorous approach is required to solve the issue with minimal artifacts and low computational complexity as a standard technology.

## 3. Proposed Spectral Compensation for the MPEG-H 3DA BR

In this section, we propose two methods of preventing spectral artifacts caused by the comb filter. Our first method, panning gain compensation (PGC), allows compensations for the panning gain of each object. As a result, it suppresses spectral notches and boosts that may occur during downmixing otherwise. PGC might be computationally heavy, as each object's left and right ear signals must be compensated separately. Our second method, binaural spectral compensation (BSC), is proposed to reduce the computational complexity of PGC. Then, two compensation methods are used in different QMF bands to compromise the system complexity with the performance.

### 3.1. Panning Gain Compensation (PGC)

Ideally, the transfer function between sound objects and the listener's ear is considered distortionless if it is equal to the true HRTF corresponding to the virtual loudspeaker position. However, it is practical to assume that the BR system has HRTFs only at sparse locations according to the pre-defined loudspeaker layout. In such cases, it is still possible to approximate the true HRTFs using geometric interpolation. A previous study [15] showed that the HRTF magnitude of a target location could be estimated via interpolation of the magnitudes of neighboring HRTFs surrounding that location, and the interpolation weights could be approximated by the 3D VBAP when the virtual source was in the far-field. Inspired by this, we approximate the magnitude of the ideal HRTF:

$$|H^j_{Target}(k)| \approx \sqrt[p]{\sum_{i=1}^{N_{ch}} \left| \tilde{g}_i H^j_i(k) \right|^p} = \| \tilde{\mathbf{g}} \odot \mathbf{H}^j(k) \|_p, \quad j = L, R, \tag{4}$$

where $\odot$ denotes element-wise multiplication. It is important to note that the norm order $p$ in Equations (1) and (4) should be identical to obtain a smooth interpolation. To validate the magnitude approximation of Equation (4), we measure the HRTF magnitude of the 5th subband of the 64-band QMF in MPEG-H 3DA operating on a 22.2-channel loudspeaker layout [16]. The results obtained along the azimuth angle $0°\sim180°$ for a fixed elevation angle at $0°$ are plotted in Figure 4. Since the 22.2-channel system has more loudspeakers on the frontal hemisphere than on the back hemisphere, the approximation accuracy is expected to be higher in the frontal region, i.e., $0°\sim90°$. On the norm order, in our experiments, we choose $p = 2$ since it provides a smoother approximation of the target HRTF than any other values as shown in Figure 4.

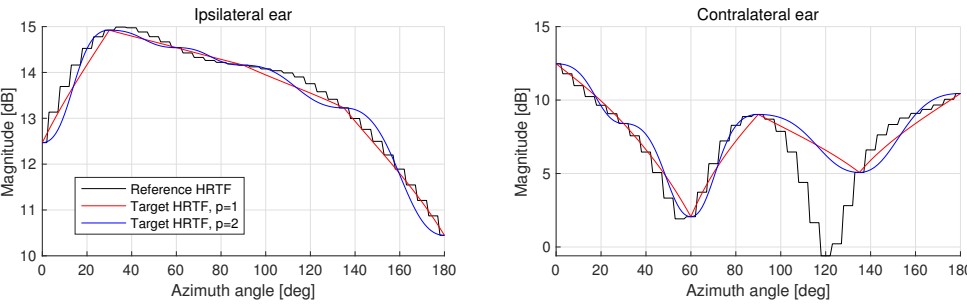

**Figure 4.** Approximation results of the HRTF magnitude in Equation (4) calculated at the 5th subband of the 64-band QMF operating on a 22.2-channel virtual loudspeaker layout. Loudspeakers are located at azimuth angles $0°$, $\pm30°$, $\pm60°$, $\pm90°$, $\pm135°$, and $180°$ in the horizontal plane.

We also design real-numbered compensation coefficients to prevent the comb filter effect. These coefficients are applied to the panning gains. Our goal is to define the real-number coefficient $\alpha^j(k)$ that satisfies the following condition:

$$\alpha^j(k)|H^j_{BR}(k)| = |\hat{H}^j_{Target}(k)|, \tag{5}$$

It simply means the panning gains are determined by computing $\alpha^j(k) = |\hat{H}^j_{Target}(k)| / |H^j_{BR}(k)|$. However, with further consideration toward the practical implementation of PGC on MPEG-H 3DA BR working in QMF domain, we calculate the compensation coefficient per QMF subband. Using Equations (3) and (4), we calculate the compensation coefficients of the subband $b$ as

$$\hat{\alpha}^j(b) = \frac{\sum_{k=k_b}^{k_{b+1}-1} \|\tilde{\mathbf{g}} \odot \mathbf{H}^j(k)\|_p}{\sum_{k=k_b}^{k_{b+1}-1} |\tilde{\mathbf{g}}^T \mathbf{H}^j(k)|}, \ j \in \{L, R\}, \tag{6}$$

where $k_b$ is the lowest fast Fourier transform (FFT) bin belongs to the frequency range of $b$-th QMF subband.

Equivalently, based on Equation (3), we apply the compensation gains directly to the panning gains as

$$\hat{g}^j_i(k) = \hat{\alpha}^j(b)\,\tilde{g}_i, \ k \in \{k_b, ..., k_{b+1} - 1\}, \ j \in \{L, R\}. \tag{7}$$

Finally, channel signals for each left and right ear are calculated as

$$\hat{X}^j_i(k) = \hat{g}^j_i(k)S(k), \ j \in \{L, R\}. \tag{8}$$

The compensation of $|H^j_{BR}(b)|$ using $\alpha^j(b)$ Equation (5) means the loudness of each subband in the downmixed signal is restored to that of the target signal. In other words, the comb filter effect is prevented by the proposed compensation of $|H^j_{BR}(b)|$.

### 3.2. Binaural Spectral Compensation (BSC)

In the previous section, PGC was proposed as an effective solution to prevent the comb filter effect. However, during PGC, every panning gain of a channel signal needs to be compensated separately for the left and right ears. This results in doubling the convolution operation as in Equation (8).

An alternative approach is binaural spectral compensation (BSC), where the down-mixed signal is compensated directly as follows.

$$|\hat{Y}^j(k)| = \beta^j(k)\left|Y^j(k)\right|, \tag{9}$$

where $\beta^j(k)$ is a real-valued gain designed for compensating the spectral notches and boosts due to the comb filter effect. The downmixed ear signal is given by

$$|Y^j(k)| = \left|(\mathbf{X}^j(k))^T\mathbf{H}^j(k)\right|, \tag{10}$$

where $\mathbf{X}^j(k) = [\, X_1^j(k), ..., X_{N_{ch}}^j(k)\,]^T$ denotes a vector comprising the channel signals. Additionally, using Equation (4), the ideal downmixed signal without spectral artifacts is obtained as

$$|\hat{Y}_{Target}^j(k)| = \left\|\tilde{\mathbf{g}} \odot \mathbf{H}^j(k)\right\|_p |S(k)| = \left\|\mathbf{X}^j(k) \odot \mathbf{H}^j(k)\right\|_p. \tag{11}$$

Additionally, to reduce unnecessary temporal fluctuation, it is possible to obtain the compensation gain for the QMF subband $b$ of the MPEG-H 3DA BR as

$$\hat{\beta}^j(b) = \frac{A\left\{\sum\limits_{k=k_b}^{k_{b+1}-1}\left\|\mathbf{X}^j(k) \odot \mathbf{H}^j(k)\right\|_p\right\}}{A\left\{\sum\limits_{k=k_b}^{k_{b+1}-1}\left|(\mathbf{X}^j(k))^T\mathbf{H}^j(k)\right|\right\}}, \tag{12}$$

where $A\{\cdot\}$ denotes a time-smoothing operator conveniently implemented using a 1st-order IIR recursive filter.

Finally, a spectrally compensated binaural signal is obtained as

$$|\hat{Y}^j(k)| = \hat{\beta}^j(b)|Y^j(k)|, \; k \in \{k_b, ..., k_{b+1} - 1\}. \tag{13}$$

From the implementation perspective, calculating a non-integer norm is highly complex. Therefore, we use the norm order $p = 2$ for ease of implementation, which was also validated from the performance perspective in the previous section. However, even with $p = 2$, calculating the square root for each frequency bin $k$ is a significant burden for the rendering processor. To circumvent this problem, we further modify the numerator and denominator of Equation (12) as

$$A\left\{\sum_{k \in b}\left\|\mathbf{X}^j(k) \odot \mathbf{H}^j(k)\right\|_2\right\} \approx A\left\{\sqrt{\sum_{k \in b}\|\mathbf{X}^j(k) \odot \mathbf{H}^j(k)\|_2^2}\right\}, \tag{14}$$

$$A\left\{\sum_{k \in b}\left|(\mathbf{X}^j(k))^T\mathbf{H}^j(k)\right|\right\} \approx A\left\{\sqrt{\sum_{k \in b}\left|(\mathbf{X}^j(k))^T\mathbf{H}^j(k)\right|^2}\right\}. \tag{15}$$

Therefore, the square root is computed per QMF band $b$, significantly reducing computational complexity. In contrast, our experiments show that the computational accuracy remains within 95% of the original value, as long as a sufficient number of frequency bins ($>10$) are included.

*3.3. Combination of PGC and BSC*

The downside of BSC is the discrepancy between the rendered signal and the target signal when there exists more than one acoustic object in a single band. In those cases ($m > 1$), the BSC gain $\hat{\beta}^j(b)$ is likely to be incorrect for every object. Especially, when two objects exist in different critical bands belonging to the same QMF band $b$, experiments show that artifacts can be audible. Therefore, although the BSC is computationally much simpler than the PGC, it is applied only to QMF bands with a comparable or narrower bandwidth than the critical bandwidth in our implementation.

It can be noted that, when $p = 2$, the BSC gain in Equation (12) is equivalent to the active downmixing gain in [11]. The purpose of developing the BSC in this paper, however, is to compromise between the increase in computational complexity and the deterioration of the sound quality of MPEG-H 3DA BR within a unified framework comprising two different compensation strategies.

## 4. Implementation and Complexity

The proposed algorithm was implemented on the frequency–domain binaural rendering of the MPEG-H 3DA reference software. Figure 5 shows block diagrams of the MPEG-H 3DA BR comprising the PGC and BSC blocks. As illustrated, the PGC is applied to the output of object render (amplitude panning) before the binaural rendering block, while the BSC is applied to the downmixed binaural signal. In the figure, $r$ denotes the QMF time slot, and $l$ and $r'$ denote the index of the short-time segmented frame and the QMF time slot index of the segmented frame, respectively, i.e., $r = l * N_{seg} + r'$, where $N_{seg}$ is the size of the short-time frame. In pursuit of the implementation, the compensated panning gains for the PGC, i.e., $\hat{g}_i^j(k)$ in Equation (7), are calculated for all azimuth and elevation angles at 1-degree intervals and stored in a lookup table. The system was designed to refresh the panning gains at every time slot of the QMF subband.

The PGC and BSC are selectively used in different QMF bands according to the bandwidth of the critical band and processing band, i.e., QMF bandwidth, as described in Section 3.3. The bands below 6 kHz are compensated using the PGC. For the bands below 750 Hz, PGC gains of the left and right ears are averaged to use for both ear signals. This is because the phase difference between the left and right ear's HRTFs is relatively insignificant. The BSC is employed for the bands above 6 kHz, where the critical bandwidth is wide enough to cover multiple QMF bands. To avoid artifacts, we limit the magnitude of BSC gain to have values within $-4.8$ dB$\sim$4.8 dB, and $A\{\cdot\}$ is implemented using a 1st-order IIR low-pass filter with a time constant of 10 ms.

Tables 1 and 2 summarize the computational complexities of the PGC and BSC blocks in the FD-BR in a unit of MOPS (million operations per second). In the tables, $N_{obj}$ is the number of object signals as in Equation (1), and $N_{pann}$ is the averaged number of loudspeakers involved in the amplitude panning, i.e., $N_{pann} = 3$ for 3D VBAP. $N_{ch}$ is the total number of virtual loudspeakers, and $N_{fft} = 2 * N_{seg}$ is the FFT size. Weight in the rightmost column of the tables is based on [17] and $F$ accounts for the number of real operations of the butterfly in the FFT algorithm, i.e., $F = 2.5$ [13]. $N_{band}$ is the number of subbands over which the FD-BR is performed. $N_{frm}$ is the number of FFT frames per second, calculated as $N_{frm} = (F_s/64)/N_{seg}$, where $F_s$ is the sampling frequency.

Using Tables 1 and 2 [13], the worst-case MOPS of the MPEG-H 3DA BR can be counted using HRTFs as a virtual loudspeaker, with and without the proposed compensation method. For the calculation, we choose MPEG-H 3DA low complexity (LC) profile level 4, which is a scenario of virtual loudspeaker rendering of 28 objects on the 22.2 channels, i.e., $N_{ch} = 22$, at a sampling rate of $Fs = 48$ kHz. We set the parameters considering a typical operating condition of FD-BR using HRTFs: $N_{fft} = 16, N_{band} = 48, N_{frm} = 93.75$, and $F_s = 48,000$. The counted MOPS of the FD-BR are 71.4 and 61.4, respectively, for the cases with and without the proposed compensation algorithm.

We can estimate the contribution of the proposed method to the total computation cost of the MPEG-H 3DA decoding and binaural rendering process. The numbers are

obtained under the condition that each QMF band selectively employs either PGC or BSC, as explained previously. Considering that, the proposed compensation algorithm takes about 3.1% and 16.3% of the worst-case complexity of the entire MPEG-H 3DA decoding/rendering process [1] and the MPEG-H 3DA BR, respectively. This means that the increase in complexity by employing the proposed method is marginal. In fact, the computational growth of 10 MOPS implies that the proposed method is affordable, even for low-power mobile devices.

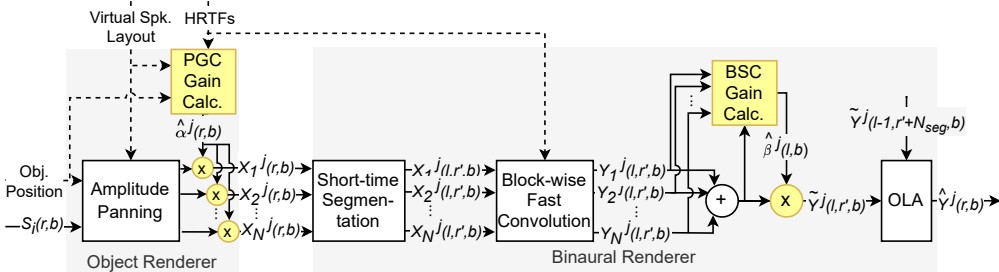

**Figure 5.** Block diagram of the MPEG-H 3DA BR comprising the PGC and BSC.

**Table 1.** Computational complexity of the PGC implemented on the MPEG-H 3DA BR.

| Processing | Operation Type | Operation Count | Weight [17] |
|---|---|---|---|
| Panning gain applying | Mult. and Add. | $2(F_s/64)N_{band}$ $\cdot N_{obj}N_{pann}$ | 1 |
| Channel signal FFT | FFT | $N_{frm}N_{ch}N_{band}$ | $F \cdot N_{fft}$ $\cdot log_2 N_{fft}$ |

**Table 2.** Computational complexity of the BSC implemented on the MPEG-H 3DA BR.

| Processing | Operation Type | Operation Count | Weight [17] |
|---|---|---|---|
| Sum of power calculation | Mult. and Add. | $4N_{frm}N_{fft}$ $\cdot N_{band}N_{ch}$ | 1 |
| Power of sum calculation | Mult. and Add. | $4N_{frm}N_{fft}$ $\cdot N_{band}$ | 1 |
| Binaural DMX gain calculation | Div. | $2N_{frm}N_{band}$ | 18 |
| | Sqrt. | $2N_{frm}N_{band}$ | 10 |
| Gain threshold | Add. and Branch | $8N_{frm}N_{band}$ | 5 |
| One-pole gain smoothing | Mult. and Add. | $2N_{frm}N_{band}$ | 1 |
| Binaural DMX gain applying | Mult. | $4N_{frm}N_{band}$ | 1 |

## 5. Experimental Results

### 5.1. Distortion Measurements

In this section, the objective performance of the PGC and BSC, respectively, is compared. Then, performance improvement of the proposed algorithm in Section 4 on MPEG-H 3DA BR is measured. To assess the performance of the proposed algorithm, we measure the spectral distortion (SD) and the interaural level difference (ILD) error. SD is calculated as a sum of the root-mean-square difference between the compensated transfer function and the actual HRTF. The measurements are conducted under the virtual 22.2-channel loudspeaker configuration using four HRTF dataset, the diffuse-field equalized (DFE) MIT HRTF [14] with large and normal pinnae and free-field equalized (FFE) CIPIC HRTF [18] with large and small pinnae. Both the SD and ILD are measured over the entire upper hemisphere using all available HRTFs in each dataset:

$$SD(b) = 20 \log_{10} \sqrt{\frac{1}{2 \cdot N_b} \sum_{j=L,R} \sum_{k \in K_b} \left( |\hat{H}_{BR}^j(k)| - |H_{ref}^j(k)| \right)^2}, \qquad (16)$$

where $K_b$ is a set of FFT bins belonging to the $b$th band and $N_b$ is the number of bins in the set $K_b$. The ILD error is measured as a difference between ILDs for the true and compensated HRTF cases using a definition of ILD: $ILD(b) = 10 \log_{10}(\sum_{k \in K_b} |H^L(k)|^2 / \sum_{k \in K_b} |H^R(k)|^2)$, where $H_L(k)$ and $H^R(k)$ are the left and right ear HRTFs, respectively. Since there is no trend difference for the different HRTF dataset in all experiments, the SD and ILD error are averaged for all HRTF dataset.

Table 3 shows the SD measured in an octave band scale of MPEG-H 3DA BR, and MPEG-H 3DA BR with the PGC and BSC, respectively. The frequency band marked with an asterisk indicates that the confidence intervals between the PGC and BSC do not overlap. The SD improvement of the PGC is 0.2–0.5 dB higher than that of the BSC in the 1.5–12 kHz frequency band, since the compensation gain of the BSC is limited. For the same reason, the standard deviation of the PGC is also smaller than that of the BSC in the 0.75–6 kHz frequency band. Thus, it is verified that the PGC compensates spectral artifacts more effectively than the BSC. Note that there is no difference in an octave band ILD error between the two methods.

**Table 3.** Mean and standard deviation of the SD measurement of MPEG-H 3DA BR, and MPEG-H 3DA BR with PGC and BSC, respectively.

| Freq. Range | 0–0.75 kHz * | 0.75–1.5 kHz * | 1.5–3 kHz * | 3–6 kHz * | 6–12 kHz * | 12–18 kHz |
|---|---|---|---|---|---|---|
| MPEG-H 3DA BR | 2.91 ($\pm$0.95) dB | 2.47 ($\pm$1.27) dB | 2.95 ($\pm$2.34) dB | 3.90 ($\pm$2.66) dB | 4.77 ($\pm$2.28) dB | 5.57 ($\pm$2.52) dB |
| MPEG-H 3DA BR with BSC | 0.44 ($\pm$0.25) dB | 0.90 ($\pm$0.71) dB | 1.84 ($\pm$1.70) dB | 2.40 ($\pm$1.93) dB | 3.91 ($\pm$2.01) dB | 4.85 ($\pm$2.32) dB |
| MPEG-H 3DA BR with PGC | 0.46 ($\pm$0.26) dB | 0.86 ($\pm$0.58) dB | 1.62 ($\pm$1.30) dB | 1.87 ($\pm$1.17) dB | 3.72 ($\pm$1.96) dB | 4.79 ($\pm$2.36) dB |

* Frequency band where the confidence intervals between the PGC and BSC do not overlap.

The SD and ILD errors of the MPEG-H 3DA BR with and without the proposed algorithm described in Section 4 are shown in Figure 6a,b. In Figure 6a, the proposed algorithm improves SD by about 2.5 dB below 750 Hz. This low-frequency range is where spectral boosts often occur since the phase difference between HRTFs is negligible. In the frequency range from 3 kHz to 12 kHz, where the spectral notch and boost simultaneously occur, the SD improvement decreases to 0.8~2 dB on average. The trend slightly changes as the frequency increases; however, the proposed method clearly reduces SD in all frequency bands.

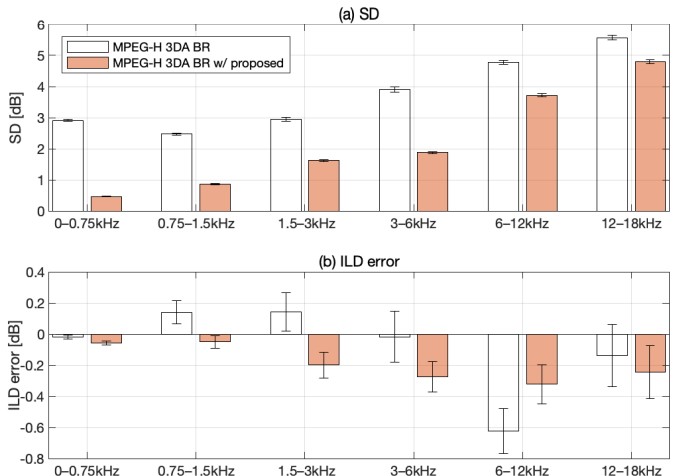

**Figure 6.** Measurement results of (**a**) SD and (**b**) ILD error with a 95% confidence interval, measured in octave bands.

In Figure 6b, the proposed algorithm maintains the ILD error within 0.5 dB in all octave bands. This means that after applying the proposed algorithm, the perceptual error becomes negligible, as it is below the just noticeable difference (JND) [19].

Based on these improvements, we argue that the compensation scheme of the proposed algorithm successfully reduces the spectral artifacts by the comb filter effect while preserving the localization of the original rendering.

### 5.2. Subject Evaluation

A MUSHRA (multiple stimuli with hidden reference and anchor) [20] test was conducted in a 3DoF rendering situation. We used the official 22.2-channel layout test materials, and seven signals included in MPEG-H 3DA call for proposal (CfP) [21]. They include recorded and synthesized 22.2-channel signals as explained in the Appendix. All binaural signals were generated using the DFE MIT HRTF [14] with normal pinnae. In the simulation of the 3DoF rendering, each channel of the test materials was regarded as an object signal located at the corresponding virtual loudspeaker position. The yaw, pitch, and roll angles of the simulated head rotation were $15°$, $0°$ and $10°$, respectively.

In the MUSHRA test, five systems were evaluated: the original MPEG-H 3DA 3DoF rendering system ("MPEG-H 3DA BR"), the same system employing the proposed algorithm ("MEPG-H 3DA BR w/ Proposed"), a hidden reference ("Ref"), and two anchors ("Anc1" and "Anc2"). The hidden reference was obtained by filtering the object signal using the actual HRTF nearest to the panning direction. The two anchors were 3.5 kHz and 7 kHz low-pass filtered versions of the hidden reference. Fifteen experienced subjects aged from 23 to 45 participated in the test.

The test results are plotted in Figure 7. We can see that the proposed algorithm significantly improves the subjective quality for all the items. The averaged MUSHRA scores of "MPEG-H BR" and "MPEG-H BR w/Proposed" are 66.1 and 87.1, respectively, with a statistically significant difference by a large margin. After the experiments, we also collected some feedback about the proposed system. Many participants reported that dullness and coloration problems were greatly alleviated by the proposed algorithm.

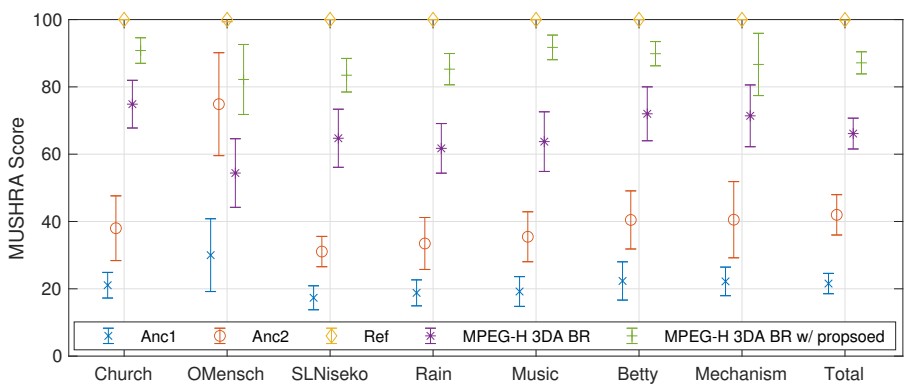

**Figure 7.** The MUSHRA test results.

### 6. Conclusions

In this paper, we proposed an effective algorithm to improve the sound quality of MPEG-H 3DA binaural rendering. The goal of our algorithm is to reduce the spectral artifacts caused by the comb filter effect. In low-frequency bands, the proposed algorithm allows independent compensation of the panning gain for each object at a different position. The panning gains are adjusted with the consideration of avoiding spectral notches and boosts occurrence during downmixing. In high-frequency bands, the proposed algorithm uses a more straightforward method of compensating the downmixed binaural signal. It provides a computationally affordable solution while preserving the sound quality in both timbral and spatial aspects. In the experiment, the proposed algorithm was implemented on the FD-BR

of the MPEG-H 3DA reference software. We could confirm empirically that the additional complexity of the proposed algorithm was minor, allowing it to be used even on low-power mobile devices. Finally, objective and subjective tests showed that the proposed method achieves the goal—improving the subjective quality of MPEG-H 3DA BR by reducing the spectral distortions significantly. In the future, we intend to extend the proposed method to the case where the transfer function of a virtual loudspeaker is BRIR. This will improve binaural audio experience even further by taking early reflection and late reverberation into consideration.

**Author Contributions:** Conceptualization, H.M. and Y.-c.P.; methodology, H.M.; software, H.M.; validation, H.M. and Y.-c.P.; formal analysis, H.M. and Y.-c.P.; investigation, H.M. and Y.-c.P.; resources, H.M.; data curation, H.M.; writing—original draft preparation, H.M.; writing—review and editing, H.M. and Y.-c.P.; visualization, H.M.; supervision, Y.-c.P.; project administration, Y.-c.P.; funding acquisition, H.M. and Y.-c.P. All authors have read and agreed to the published version of the manuscript.

**Funding:** This work was supported by the Technology Innovation Program (10080057, MPEG-I Binaural Audio Codec Standards for Immersive Virtual Reality Contents) funded by the Ministry of Trade, Industry & Energy (MOTIE, Korea).

**Conflicts of Interest:** The authors declare no conflict of interest.

## Appendix A

This appendix introduces the 22.2-channel layout listening test materials used in Section 5.2. Seven test materials of MPEG-H 3DA CfP [21] are used to generate the 22.2-channel test material for the MUSHRA test. There are three recorded-type 22.2-channel audio ("Church", "OMensch", "SLNiseko") and four mixed-type audio, i.e., an audio scene composed of recorded and synthesized audio ("Rain", "Music", "Betty", and "Mechanism"). Detailed information [13] is shown in Table A1. The sound object is rendered to the 22.2 channel as in the MPEG-H CfP binaural listening test [21].

**Table A1.** List of test items.

| Name | Description | Duration [Seconds] | Format | Type |
|------|-------------|--------------------|--------|------|
| Church | Multiple bell sounds with various pitch | 16 | 22.2 Ch. | Recording |
| OMensh | Pipe organ music, J. S. Bach's "Toccata and Fugue in D minor, BVW 565" | 27.4 | 22.2 Ch. | Recording |
| SLNiseko | A sound of a passing steam locomotive | 19 | 22.2 Ch. | Recording |
| Rain | A single audio object that are moving foot steps and 22.2-channel rain sound as background | 16 | 22.2 Ch. 1 Obj. | Mix |
| Music | One dry vocal object with 5.0 channel vocal reverb and 22.2-channel instrument sounds (latin music) | 20.7 | 22.2 Ch. 6 Obj. | Mix |
| Betty | "Message in the Snow" from Betty Lenard | 10.4 | 30 Obj. | Mix |
| Mechanism | Several industrial and mechanical sounds (motor, impact, steam and other metal related sounds) | 9 | 30 Obj. | Mix |

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
