# Peer review of "Quality Enhancement of MPEG-H 3DA Binaural Rendering Using a Spectral Compensation Technique"

_electronics, doi:10.3390/electronics11091491_

Round 1

Reviewer 1 Report

The paper presents an efficient spectral compensation algorithm for mitigating spectral artifacts due to the comb filter effect. The proposed algorithm  performs spectral compensation in both the panning gain and downmix signal domains. It was implemented on the MPEG-H 3DA BR to assess its computational complexity, and its performance was confirmed through objective and subjective tests.

The paper is written very well. The topic addressed has been introduced properly and the proposed method has been described clearly. The results presented corroborate the proposed method.

In the simulations section, would it be possible to add a comparison with some concurrent methods?

Minor issues:

Section 2: "the associated metadata is processed..." Data is a plural noun, so data are processed.

Section 3.1: "where kb is the lowest fast Fourier transform (FFT) bin belongs to the QMF subband b." - bin that belongs??

Section 4: "Figs. 5 shows" - Fig. 5 shows.

In Fig. 5, arrows below the PGC Gain. Calc. block should be separated a bit Consider reducing the head size.

Author Response

We appreciate your careful review. We attached our response to your review and revised manuscript.

Reviewer 2 Report

A very well written and clear paper. The problem is put across properly and their solution is explained in a very understandable way. The results to justify their answer are straightforward and are clear to grasp. The maths is developed properly and cleanly. The conclusions are straighforward. What is missing is a few sentences of future work. Also, some small text changes that are needed are highlighted in the annotated copy attached.

Author Response

We appreciate your careful review and feedback. We attached our response to your review and revised manuscript.
